# AKT Mediates Adiponectin-Dependent Regulation of VSMC Phenotype

**DOI:** 10.3390/cells12202493

**Published:** 2023-10-20

**Authors:** Abigail E. Cullen, Ann M. Centner, Riley Deitado, Ahmed Ismaeel, Panagiotis Koutakis, Judy Muller-Delp, Gloria Salazar

**Affiliations:** 1Department of Nutrition and Integrative Physiology, Florida State University, Tallahassee, FL 32306, USA; acullen@uoregon.edu (A.E.C.); ann.centner@med.fsu.edu (A.M.C.); rdeitado@fsu.edu (R.D.); 2Department of Human Physiology, University of Oregon, Eugene, OR 97403, USA; 3Department of Physiology, University of Kentucky, Lexington, KY 40536, USA; ahmed.ismaeel@uky.edu; 4Department of Biology, Baylor University, Waco, TX 76706, USA; panagiotis_koutakis@baylor.edu; 5Department of Biological Sciences, Florida State University, Tallahassee, FL 32306, USA; judy.delp@med.fsu.edu; 6Center for Advancing Exercise and Nutrition Research on Aging (CAENRA), Florida State University, Tallahassee, FL 32306, USA

**Keywords:** atherosclerosis, adiponectin, sex, vascular smooth muscle cells, AKT

## Abstract

Adiponectin (adipoq), the most abundant hormone in circulation, has many beneficial effects on the cardiovascular system, in part by preserving the contractile phenotype of vascular smooth muscle cells (VSMCs). However, the lack of adiponectin or its receptor and treatment with recombinant adiponectin have shown contradictory effects on plaque in mice. RNA sequence of *Adipoq^+/+^* and *adipoq^−/−^* VSMCs from male aortas identified a critical role for adiponectin in AKT signaling, the extracellular matrix (ECM), and TGF-β signaling. Upregulation of AKT activity mediated proliferation and migration of *adipoq^−/−^* cells. Activation of AMPK with metformin or AdipoRon reduced AKT-dependent proliferation and migration of *adipoq^−/−^* cells but did not improve the expression of contractile genes. Adiponectin deficiency impaired oxidative phosphorylation (OXPHOS), increased expression of glycolytic enzymes, and elevated mitochondrial reactive oxygen species (ROS) (superoxide, and hydrogen peroxide). Anti-atherogenic mechanisms targeted the ECM in *adipoq^−/−^* cells, downregulating MMP2 and 9 and upregulating decorin (DCN) and elastin (ELN). In vivo, the main sex differences in protein expression in aortas involved a more robust upregulation of MMP3 in females than males. Females also showed a reduction in DCN, which was not affected in males. Our study uncovered the AKT/MAPK/TGF-β network as a central regulator of VSMC phenotype.

## 1. Introduction

The hormone adiponectin has many protective effects on the cardiovascular system, acting as an anti-atherogenic and anti-insulin resistance modulator [1]. Low adiponectin levels can predict restenosis [2] and are seen in obese individuals [3]. Adiponectin production and secretion have historically been associated with adipose tissue; however, a multitude of tissues and cells, including skeletal and VSMCs, placental tissue, and cardiac cells, can also secrete adipokine. Still, adipose tissue is the major source of adiponectin secretion [4,5]. Adiponectin binds to several receptors, including adiponectin R1 and R2 (AR1 and AR2, respectively) and T-cadherin. AR1 and AR2 play a major role in VSMCs’ contractile phenotype, while T-cadherin plays a more important role in angiogenesis and revascularization [6]. Adiponectin mediates its effects on the metabolic state of the cell through activation of the AMP-activated protein kinase (AMPK), mitogen-activated protein kinase (MAPK), and protein kinase B (AKT) pathways [7,8]. In the vascular system, adiponectin acts in a paracrine manner, inducing the expression of contractile proteins and leading to the differentiation of VSMCs through AMPK activation [5,9]. The adiponectin receptor agonist AdipoRon was shown to inhibit PDGF-induced VSMC proliferation by an mTOR-dependent and AMPK-independent mechanism [10]. Although many molecules have been identified regulating adiponectin signaling in VSMCs, the precise molecular mechanism by which adiponectin regulates the proliferation, migration, and dedifferentiation of VSMCs is incompletely understood. Such mechanisms are relevant because they are all pro-atherogenic phenotypes of VSMCs. The de-differentiation of VSMCs also promotes neointima formation and hypertension.

Thus, adiponectin promotes a contractile/anti-atherogenic phenotype in VSMCs. However, in vivo adiponectin deficiency in low-density lipoprotein receptor (LDLR) *ldlr^−/−^* or *apoe^−/−^* mice did not affect plaque [11], suggesting that compensatory mechanisms may protect the cardiovascular system in the absence of adiponectin. In this study, we uncovered the molecular network regulated by adiponectin in VSMCs using an RNA sequence of *Adipoq^+/+^* and *adipoq^−/−^* VSMCs. Our data revealed a prominent role of adiponectin in the PI3K/AKT pathway, which was upregulated by adiponectin deficiency, inducing VSMC proliferation and migration. We also showed new insights into how adiponectin can regulate the VSMC phenotype through different mechanisms. Adiponectin plays a role in the regulation of the extracellular matrix (ECM) through the regulation of matrix metalloproteinases (MMPs) and ECM components such as collagen and transforming growth factor-β (TGF-β). A compensatory increase in AMPK activity was observed in the aortas of male and female *adipoq^−/−^* mice. Female aortas showed a robust upregulation in MMP3 and downregulation in DCN, suggesting that females lacking adiponectin may be more susceptible to atherosclerosis. 

## 2. Materials and Methods

### 2.1. Animal Model

Single knockout mice adiponectin (B6.129-Adipoqtm1Chan/J, strain # 8195) were purchased from Jackson Laboratory and housed in a 12:12 light: dark cycle with water and food provided ad libitum. Animal procedures were approved by the Institutional Animal Care and Use Committee at Florida State University. Two-month-old male mice were used to isolate VSMCs from the aorta and three- to four-month-old male and female mice (n = 8/group) were used for the analysis of protein expression by Western blot.

### 2.2. Preparation of Mouse VSMCs

VSMCs were harvested from the aortas of male C57Bl/6 *Adipoq^+/+^* and *adipoq^−/−^* mice using an explant method, as reported [12]. The aortas were clean of periadventitial fat and treated with collagenase (150 U/mL, Worthington, Lakewood, NJ, USA; 4202) for 30 min at 37 °C to remove the tunica adventitia. The aorta was then cut lengthwise, and the intima was removed with a sterile cotton swab. The remaining media layer was cut into small pieces and allowed to attach to the culture plate for at least 3 days. Explants were cultured until plates were 80% confluent. Cells were expanded, frozen using 10% DMSO (Sigma-Aldrich, St. Louis, MO, USA; 1001731596) and 90% FBS (Seradigm from Avantor, Allentown, PA, USA; 97068-075), and stored in liquid nitrogen. Cells were used for experiments up to passage 12.

### 2.3. Cell Culture

Cells were grown in T75 cell culture flasks (Greiner, Monroe, NC, USA; 658170) and cultured in Dulbecco’s modified Eagle medium (DMEM) (Corning, Corning, NY, USA; 10-014-CV) supplemented with 10% FBS, glutamine (2 mM), penicillin (100 U/L), and streptomycin (100 mg/mL) (Corning, Corning, NY, USA; 30-009-CI). Cells were cultured at 37 °C in a 5% CO_2_ incubator, and the medium was changed every other day. For treatment, cells were starved in DMEM with 0.2% FBS for 24 h before subsequent treatments.

### 2.4. Cell Lysis and Sample Preparation

After treatment, plates were placed on ice and washed twice with cold PBS containing calcium and magnesium (PBS Ca/Mg) (1 mM MgCl_2_, 0.1 mM CaCl_2_) and were lysed with 120 µL of Buffer A (50 mM HEPES, pH 7.4, 150 mM NaCl, 1 mM EGTA and 0.1 mM MgCl_2_) containing 1% Triton X-100 (Sigma-Aldrich, X100-500ML), 2 mM sodium orthovanadate (Enzo Life Sciences, Farmingdale, NY, USA, 400-032-G025), 10 mM sodium pyrophosphate (Sigma-Aldrich, 221368), 10 mM sodium fluoride (J.T. Baker, Phillipsburg, NJ, USA, 368-01), and protease inhibitor cocktail (Sigma, P340-5ML). Samples were sonicated 3 times for 10 s each using a QSonica sonicator and protein was measured using Bradford Assay (BioRad, Hercules, CA, USA; 50000006) at 595 nm in a spectrophotometer. Total extracts (20–50 µg) were separated in either 4–20% acrylamide precast Criterion gels (BioRad, Hercules, CA, USA; 5671094) or homemade 8%, 10%, or 12% SDS-PAGE gels.

### 2.5. Western Blot

Gels were transferred to PVDF membrane (Thermo Scientific, Waltham, MA, USA; PI88518) using the semidry system Owl HEP-1 (Thermo Scientific, Waltham, MA, USA) and membranes were blocked for 20 min in TBS (150 mM NaCl, 2 mM KCl, 25 mM Tris, pH 7.4) containing 1.5% non-fat dry milk (Bio-Rad, Hercules, CA, USA; 170-6406). After washing in TBS, membranes were incubated with primary antibodies from 1 to 2 h to overnight. Membranes were then washed in TBS-T (TBS plus 0.05% T-X100) 3 times, 10 min each, and were incubated with HRP-conjugated secondary antibodies in a blocking buffer for 45 min. After 3 washes in TBS-T, membranes were developed using Pierce ECL Western blotting substrate (Thermo Fisher Scientific, Waltham, MA, USA).

### 2.6. Proliferation and Cell Migration Assays

Cells (5000 cells/well) were seeded in 12 well plates and cultured in media with 10% FBS for 48 h. Cells were then incubated in media with 0.2% FBS for 24 h to synchronize cell growth cycles. Cells were switched back to the 10% FBS media and counted every day in triplicate with a hemocytometer chamber for 6 d. Cell migration was assessed via scratch wound analysis. Cells were seeded in 6 well plates (with grid lines indicating where scratch wounds will occur) and grown to confluency. Cells were starved for 24 h in 0.2% FBS and then a 1 mL pipette tip was used to create the scratch wound and pictures were taken. Cells were then cultured in media with 0.2% FBS and pictures were taken after 12 and 24 h of treatment. Migration was assessed by measuring scratch wound width using Adobe Photoshop and was presented as percent reduction. The percent closure was calculated as follows: |(cm Xh − cm 0 h) × 100)/cm 0 h)|, with 0 representing the open wound at the beginning and 100% a completely closed wound. Proliferation and scratch wound analyses were performed on *Adipoq^+/+^* and *adipoq^−/−^* cells with no treatment and *adipoq^−/−^* cells treated with recombinant adiponectin (10 μg/mL), metformin (5 mM), AdipoRon (10 μM), erlotinib (Erl, epidermal growth factor receptor/EGFR inhibitor, 100 nM), triciribine (TCB, AKT inhibitor, 1 μM), PD-98059 (PD, mitogen-activated protein kinase kinase MEK/MAP2K1 inhibitor, 10 μM), and SB-252334 (SB, TGF-β receptor inhibitor, 5 μM).

### 2.7. Zymography

To assess gelatinase activity, cells were grown to 80–90% confluency, starved for 24 in DMEM 0.2% FFS, and then treated in plain media for 24 to 48 h. The media were collected and concentrated with an Amicon Ultra-0.5 column with a 10 kDa molecular weight cutoff. Samples of concentrated media were separated under non-reducing conditions and without heating in an 8% SDS-PAGE gel with 1% gelatin (Sigma-Aldrich, St. Louis, MO, USA; G7041-100G). The gel was washed in 2.5% Triton X-100 in water for 30 min, with one change of the solution after 15 min. The gels were then rinsed in substrate buffer (500 mM Tris-HCl, pH 7.8, 2 mM NaCl, 50 mM CaCl_2_) overnight and incubated in staining buffer (0.25% Coomassie blue R-250, 5% methanol, and 10% acetic acid) for 2 h. After the gel was fully dyed, it was rinsed in a destaining buffer (10% methanol, 5% acetic acid) overnight.

### 2.8. Oroboros Oxygraph System

To determine the function of the mitochondrial complexes associated with oxidative phosphorylation, we used the Oroboros oxygraph system (Oroboros Instruments, Innsbruck, Austria), as previously reported [13]. *Adipoq^+/+^* and *adipoq^−/−^* cells were grown to confluency, and approximately 1–2 million cells were resuspended in mitochondrial respiration medium MiR05 (0.5 mM EGTA, 3 mM MgCl_2_, 60 mM lactobionic acid, 20 mM taurine, 10 mM KH_2_PO_4_, 20 mM HEPES, 110 mM D-sucrose, and 1g/L fatty acid-free BSA, pH 7.1), supplemented with 20 mM creatine monohydrate. Digitonin (4.05 μM) was used to permeabilize the plasma membranes. To stimulate electron flow through Complex I, malate (2 mM) and glutamate (10 mM) were added to the chambers to produce NADH (state 2 respiration). To measure ADP-stimulated oxygen consumption by Complex I (state 3 respiration), ADP was added (4 mM) to the chambers. Next, succinate (10 mM) was added to stimulate electron flow through Complex II as well, followed by rotenone (10 μM) to inhibit Complex I, thus measuring Complex II respiration independent of Complex I. Finally, Complex IV respiration was assessed using N, N, N′, N′-Tetramethyl-p-phenylenediamine dihydrochloride (TMPD, 0.4 mM), 2 mM ascorbate to prevent TMPD auto-oxidation, and 5 μM antimycin A to inhibit electron flow through Complex III. Cytochrome c (10 μM) was added to test mitochondrial membrane integrity, and any samples with a relevant cytochrome c release were excluded. The respiration rate was expressed as pmol oxygen consumed per second and normalized to protein content in the O2k chamber (pmols/s/mg protein), measured using a Pierce BCA protein assay (Thermo Fisher Scientific).

### 2.9. Nuclear Fractionation

Cells were washed twice with PBS Ca/Mg and resuspended in 300 µL of hypotonic buffer (20 mM Tris, pH 7.4, 3 mM MgCl_2_, and 10 mM NaCl plus 10 μL/mL protease inhibitor cocktail), as previously reported [14]. Extracts were collected using a cell scraper and transferred to pre-chilled 1.7 mL microcentrifuge tubes and incubated on ice for 15 min. Then, 15 µL of 10% NP40 was added to the samples, followed by vortex for 10 s. The samples were centrifuged at 4 °C 22,000× *g* for 10 min. The supernatant was saved and labeled Cytosolic Fraction. The pellet was washed with 500 µL hypotonic buffer and the excess buffer was removed. The pellet was suspended in 80 µL of nuclear extraction buffer (100 mM Tris, pH 7.4, 100 mM NaCl, 1 mM EDTA, 10% glycerol, 1 mM EGTA, 0.1% SDS. 0.5% deoxycholate, 1% TritonX-100, and 10 μL/mL protease inhibitor cocktail) and incubated on ice for 30 min, vortexing every 10 min. Finally, the pellet samples were centrifuged for 20 min at 22,000× *g* at 4 °C. The supernatant was collected and labeled Nuclear Fraction.

### 2.10. RNA Sequence

Total RNA sample QC, RNA concentration, RIN values, 28S/18S, and fragment distribution were performed using an Agilent 2100 BioAnalyzer (Agilent RNA 6000 Nano Kit). Poly-A-containing mRNA molecules were purified using poly-T oligo-attached magnetic beads and then fragmented into small pieces using divalent cations under elevated temperatures. The RNA fragments were copied into first strand cDNA using reverse transcriptase and random primers, which was followed by a second strand cDNA synthesis using DNA Polymerase I and RNase H. cDNA fragments had the addition of a single ‘A’ base and subsequent ligation to the adapter. The products were purified and enriched by PCR amplification. The PCR yield was quantified by Qubit and samples were pooled together to produce a single-strand DNA circle (SSDNA circle), generating the final library. DNA nanoballs (DNBs) were produced with the ssDNA circle by rolling circle replication (RCR) to enlarge the fluorescence signals during the sequencing process. The DNBs were loaded into the patterned nanoarrays, and pair-end reads of 100 bp were read through on the DNBseq platform for the following data analysis study. For this step, the DNBseq platform combined the DNA nanoball-based nanoarrays and stepwise sequencing using the combinational probe–anchor synthesis sequencing method.

Sequencing reads’ filtering was performed using internal software SOAPnuke, (v1.5.2) which removed reads with adaptors, rads with unknown bases, and low-quality reads. After filtering, the clean reads were stored in FASTQ format. For gene expression analysis, clean reads were mapped using Bowtie2, and gene expression levels were calculated with the RSEM software package (v1.2.12). Pearson correlation between all samples was calculated using Cor, and PCA analysis was calculated using princomp.

DEGs were detected using DEseq2 PossionDis. DEseq2 was based on the negative binomial distribution, performed as described by Michael I. et al. [15]. PossionDis was based on the Poisson distribution, performed as described by Audic S. et al. [16].

For gene ontology (GO) and KEGG annotation, DEGs were classified according to official classification. Functional and pathway enrichment was calculated using phyper, a function of R. Upregulated and downregulated genes were identified as having a log2 transformed fold change between control and samples > and < than 0.58, respectively.

About 5.93 Gb bases were generated per sample. The average mapping ratio with reference genome was 95.40%, and the average mapping ratio with gene was 79.45%; 16,851 genes were identified, of which 16,851 were known genes and 1178 were novel genes. A total of 11,092 novel transcripts were identified, of which 8111 were previously unknown splicing events for known genes, 1178were novel coding transcripts without any known features, and the remaining 1803 were long noncoding RNA.

### 2.11. Statistics

Western blot results were analyzed using ImageJ. Protein expression was determined in at least 3 independent experiments, adjusted by the level of loading control (tubulin, GAPDH, or β-actin) and data were analyzed by comparing *Adipoq^+/+^* and *adipoq^−/−^* cells (2 groups) using two-way *t*-tests in JMPPro15 (version 15.0.0). For cell culture experiments utilizing time as a variable (proliferation curve and scratch wound assay), a mixed model repeated measures test (genotype, time, and genotype* time) was used. Values were given as mean ± standard deviation of mean (SD), except for body weight, food, and water, which were given as mean ± standard error (SE), and *p* < 0.05 was considered statistically significant.

## 3. Results

### 3.1. Adiponectin Deficiency Upregulates the AKT Signaling Pathway and Reduces VSMC Differentiation

To obtain insights into the mechanisms by which adiponectin regulates the VSMC phenotype, we isolated VSMCs from Adipoq^+/+^ and adipoq^−/−^ male mice and performed RNA sequencing analysis. The results yielded 15,986 genes, of which 1054 were upregulated and 1187 were downregulated in adipoq^−/−^ cells compared with WT (Figure 1A). Differentially expressed genes (DEGs) were labeled as up or down if the log2 of the fold change (adipoq^−/−^/Adipoq^+/+^) was higher or lower than 0.58, respectively. The most upregulated gene was the cell adhesion molecule contactin-associated protein 2 (Cntnap2, ~600-fold), while the most downregulated gene was the complement component 4 (C4b, ~270-fold), followed by fibromodulin (Fmod, ~80-fold) and Apoe (~70-fold).

Gene ontology analysis of the DEGs showed many genes were involved in cellular processes, human diseases, and metabolism (Figure 1B). For example, 387 genes were associated with signal transduction, while 236 were involved in cancer.

Further analysis of the most enriched pathways (Figure 1C) revealed prominent changes in the phosphoinositide-3-kinase (PI3K)-AKT signaling pathway, with ~40 genes upregulated and ~50 downregulated. Network analysis (Figure 1D) of prominent pathways showed close interactions among the PI3K-AKT, pathways in cancer, Rap1 signaling, and the ECM-receptor interaction networks. Many genes in the cancer network overlapped with the PI3K/AKT, as both pathways regulated cell growth.

Some of the most enriched (Kyoto Encyclopedia of Genes and Genomes) KEGG pathways (Figure 2A) affected also included the Ras and MAPK cascades, cell adhesion, focal adhesion, ECM-receptor interaction, pathways in cancer, axon guidance, and complement and coagulation cascades. Further analysis of the ECM-receptor interaction, MAPK signaling, and PI3K/AKT pathways indicated many alterations in the absence of adiponectin. The most significant changes are indicated by arrows on the heatmaps shown in Figure 2B.

Next, we focused on AKT and MAPK signaling pathways that are known to interact with AMPK, the major downstream effector of the adiponectin receptor, and compared the expression from the RNA sequencing (Appendix A) to the protein levels (Figure 2C,E). Phosphorylation of the AMPK catalytic subunit (p-PRKAA2) was reduced by 50%, while total levels were upregulated in adipoq^−/−^ cells. Consistent with reduced AMPK activity, AKT phosphorylation was strongly upregulated ~14-fold, which could be mediated by the upregulation of Akt1 mRNA. Adipoq^−/−^ cells also expressed higher levels of phosphorylated mitogen-activated protein kinase kinase 1 (MAP2K1/MEK) and its downstream kinase MAPK1/ERK. Because AKT and MAPK1 negatively impact AMPK activity [17,18], our data suggest that a negative loop mediated by these kinases may contribute to AMPK reduced activity. In addition to the upregulation of proliferative signaling pathways, protein, but not RNA expression, of the cell cycle progression kinase cyclin D1 (CCND1) was also increased by the lack of adiponectin. In addition to the AMPK pathway, activation of AR1 and AR2 receptors also stimulated p38MAPK (MAPK14) [19]. Consistent with the lack of adiponectin, MAPK14 phosphorylation was significantly reduced in the KO cells. Changes in the MAP kinases and AKT phosphorylation were not caused by changes in protein expression, as the expression of these kinases did not differ between genotypes (Appendix A and Figure 2E).

VSMCs have been shown to secrete adiponectin, which promotes a contractile phenotype in VSMCs [5]. In line with this report, adipoq^−/−^ VSMCs RNA sequence (Appendix A) and Western blot data (Figure 2D,F) showed reduced expression of contractile genes, including transgelin (TAGLN/SM22), smooth muscle actin (ACTA2), smooth muscle myosin heavy chain (MYH), and myosin light chain (MYL).

Contractile genes are regulated by the transcriptional coactivator myocardin (MYOCD) and transcription factor serum response factor (SRF) [20]. Negative regulators of MYOCD, such as Krüppel-like factor 4 (KLF4), prevent the formation of the MYOCD/SRF complex, reducing the expression of differentiation genes. Interestingly, while the differentiation markers tested were significantly reduced in adipoq^−/−^ cells, MYOCD and SRF were upregulated at the RNA (Appendix A) and protein levels (Figure 2D,F). RNA and protein levels for KLF4 were increased and reduced, respectively, in the adipoq^−/−^ cells. Next, we tested whether altered subcellular localization of these genes may explain the reduced expression of differentiation markers. Adipoq^−/−^ cells showed a robust reduction in nuclear SRF and no differences in the level of MYOCD or KLF4 in the nucleus (Figure 2G,H). Clathrin heavy chain (CLTC) and histone H3 (H3) were used as cytosolic and nuclear markers, respectively.

### 3.2. Loss of Adiponectin Promotes a Synthetic Phenotype Altering ECM Composition

Increased cell signaling of the AKT and MAPK pathways and reduced expression of contractile genes induced a switch toward a synthetic phenotype. Images of cell cultures (Appendix A) showed more mitotic profiles (telophase and cytokinesis) in adiponectin-deficient cells. Further, the logarithmic phase of growth of VSMCs was increased in the adipoq^−/−^ cells (Figure 3A), reaching a higher cell density at days 3, 4, 5, and 6 compared with Adipoq^+/+^ cells. Since both cell types reached confluency, these data suggest that adipoq^−/−^ cells may be smaller in size. To discern if migration was also increased in parallel to proliferation, in the adipoq^−/−^ cells, we measured scratch wound closure at 12 and 24 h (Figure 3B). At both time points, there was a significant increase in the % closure of the KO cells, potentially signifying differences in ECM regulation and motility regulators, like Rac2, as suggested by RNA sequence data (Appendix A).

The switch toward the synthetic phenotype in adipoq^−/−^ cells was associated with impaired oxidative phosphorylation (OXPHOS), specifically in Complexes I and IV (Figure 3C) and by upregulation of RNA expression of glycolytic enzymes (Appendix A). We confirmed the upregulation of protein expression of glycolytic enzymes, including enolase 3 (ENO3), phosphofructokinase liver type (PFKL), a Pfk1 isoform, and phosphoglycerate mutase 1 (PGAM1) (Figure 3D,E). AKT phosphorylates PFK isoforms, increasing its protein stability and proliferation rate in glioblastoma cells [21], suggesting that increased AKT activity may induce PFK expression and upregulate glycolysis in VSMCs. Similarly, lactate dehydrogenase (LDH), the enzyme that converts pyruvate into lactate, was also upregulated in adipoq^−/−^ cells, suggesting that the NAD produced in this reaction may further drive glycolysis. Consistent with the altered function of Complexes I and IV of the ETC, adipoq^−/−^ cells have higher levels of mitochondrial ROS (MitoROS), as well as superoxide (O_2_^.−^) and hydrogen peroxide (H_2_O_2_) (Figure 3F). Altogether these data suggest that adipoq^−/−^ cells use glycolysis instead of OXPHOS to fuel their high rate of proliferation and migration.

Synthetic VSMCs remodel the ECM by secreting MMPs and ECM components, stimulating migration [22]. High expression of MMPs, in particular MMP2 and MMP9, degrade collagen and ELN, weakening the arterial wall and contributing to plaque formation [23]. However, at physiological/low levels, MMPs can be beneficial because, when the ECM is not degraded, arteries can begin to stiffen [24]. To assess whether molecules involved in ECM remodeling were affected in the *adipoq^−/−^* cells, the RNA sequence was searched for MMPs (Appendix A). Genes encoding MMPs 3, 19, 25, and 28 showed the strongest change relative to Adipoq^+/+^ WT cells. MMP2 and MMP14 were expressed at the highest level based on RNA expression.

Compared with Adipoq^+/+^ cells, MMP2 was reduced by about 50% and MMP14 was not affected in adipoq^−/−^ cells. In contrast, MMPs 3, 9, and 15 and 23, 25, and 27 were upregulated. MMP9, however, was expressed at very low levels. Consistent with the RNA data, MMP2 was reduced while MMP3 was increased in the media collected from cultured adipoq^−/−^ compared with Adipoq^+/+^ cells (Figure 3G,H). Cathepsin D (CTSD) was secreted into the media and served as a loading control. Gelatinase activity measured by zymography showed bands of about 50 and 90 kDa, consistent with MMP2 and MMP9 molecular weights, that were reduced in adipoq^−/−^ cells (Figure 3I). Like MMP2, MMP9 was also reduced in the media of these cells (Figure 3J), suggesting that reduced MMP2 and MMP9 activities may be mediated by lower protein levels. MMP9 degraded elastin (ELN) [25], while MMP2 and MMP3 degraded decorin (DCN). Further, collagen type VIII (Col8) has been shown to signal through β1 integrins to upregulate MMP2 and induce migration of smooth muscle cells [25]. In line with this evidence, reduced MMP2 and MMP9 activities correlated with DCN and ELN upregulation at the RNA (Appendix A) and protein levels (Figure 3L,M). Col8 RNA, on the other hand, was increased, while its protein level was reduced.

TGF-β is a major regulator of the ECM [26] and mediates VSMC differentiation [27]. Activation of the TGF-β1 receptor stimulates pathways that initiate the production of ECM components such as collagen [28]. The altered phenotype of adipoq^−/−^ VSMCs suggests that TGF-β signaling could be impaired in these cells. Many regulators of the ECM and iron metabolism, such as membrane type 1-matrix metalloproteinase (MT1-MMP/MMP14) and transferrin receptor (TFRC), mediate TGF-β effects. MMP14 releases TGF-β from its binding to DCN in the ECM [29] and also releases TGF-β from latent transforming growth factor-β (LTBP) [30]. DCN binds to TGF-β, preventing its binding to surface receptors, thus acting as an antifibrotic molecule. TFRC, on the other hand, enables TGF-β-induced craniofacial morphogenesis [31] and TGF-β-induced renal fibrosis [30]. Evaluation of the TGF-β pathway in the RNA sequence (Appendix A) revealed upregulation of the Tfrc, transferrin (Trf), and Ltbp1; downregulation of Ltbp2 and Ltbp4; and no change in Mmmp14 genes (Appendix A) by adiponectin deficiency. However, at the protein level, both TFRC and MMP14 were upregulated (Figure 3L,M).

Analysis of TGF-β-induced signaling revealed higher phosphorylation levels of AKT and suppressor of mothers against decapentaplegic (SMAD) 2 and higher expression of TFRC and MMP14 in adipoq^−/−^ cells compared with Adipoq^+/+^ (Figure 4A,B,D–F). Phosphorylation of MAPK1 was reduced in Adipoq^+/+^ cells by TGF-β treatment, which correlated with upregulation of ACTA2. Surprisingly, TGF-β failed to reduce MAPK1 activity and upregulate ACTA2 in adipoq^−/−^ cells (Figure 4A,C,G).

We next investigated whether activation of AKT by TGF-β was mediated via transactivation of receptor tyrosine kinases like EGFR, which was also upregulated in adipoq^−/−^ cells (Figure 5A,B). Treatment with EGF showed similar responses for AKT and MAPK1 phosphorylation (Figure 5C,D) and similar levels of EGFR at the cell membrane in both genotypes (Figure 5E,F), suggesting that upregulation of AKT activity was not caused by EGFR signaling. Caveolin 1 (CAV1) expression, on the other hand, was reduced in cell extracts and at the cell surface (Figure 5E–G). Interestingly, CAV1 inhibited SMAD2-dependent TGF-β signaling [32], suggesting that downregulation of CAV1 may promote the higher SMAD2 signaling observed in adipoq^−/−^ cells. CTLC, which is required for EGFR internalization, was unaffected by the lack of adiponectin.

These alterations suggest that the ECM surrounding adipoq^−/−^ cells is structurally different from Adipoq^+/+^ ECM, which in turn changes the rate of migration, proliferation, and differentiation of the adipoq^−/−^ cells. Altogether, these data suggest that DCN may serve as a reservoir for TGF-β, which can subsequently be released from the ECM by MMP14 and interact with the cell surface receptors to activate AKT and induce proliferation.

### 3.3. AMPK Inhibits AKT-Dependent Cell Signaling, Proliferation, and Migration but Does Not Reverse the Dedifferentiation of VSMCs

PI3K/AKT is one of the most prominent pathways regulated by adiponectin. This pathway seemed to be activated in the absence of adiponectin, as phosphorylated AKT was upregulated ~14-fold in adipoq^−/−^ cells. To investigate if increased phosphorylated AKT correlated with increased activity, we examined the activation/phosphorylation of downstream targets. The mammalian target of rapamycin (mTOR) is phosphorylated in Ser 2448 by the PI3/AKT pathway. Active mTOR Complex I (mTORC1) then phosphorylates p70 S6 kinase in Thr 389. Consistent with increased AKT phosphorylation, the phosphorylation of mTOR in Ser 2448 and p70 S6K in Thr 389 was upregulated in adiponectin-deficient cells compared with Adipoq^+/+^ cells (Figure 6A,B). Then, AKT and MAPK1 signaling was inhibited with TCB and PD, respectively, in adipoq^−/−^ cells.

Next, to determine the impact of AKT as well as MAPK signaling in the phenotypic switch, these kinases were inhibited with TCB and PD, respectively. TCB and PD alone or in combination reduced CCND1 and elevated ACTA2 (Figure 6C,G,H), as well as TAGLN (Appendix A). These data suggest that upregulation of both AKT and the MAPK pathways promotes the synthetic phenotype of adipoq^−/−^ cells. Inhibition of the EGFR by Erl showed a weaker effect in AKT and a prominent reduction in MAPK1 phosphorylation and failed to reduce CCND1 and increase ACTA2 (Figure 6D–H). Inhibition of TGF-β signaling by SB reduced all these markers (Figure 6D–H). Thus, inhibition of AKT and MAPK1 and activation of the TGF-β pathways are likely needed to maintain the contractile phenotype of VSMCs.

Next, we investigated whether activation of adiponectin signaling can downregulate AKT and MAPK1 pathways and reverse the phenotypic switch of the adipoq^−/−^ cells. We tested this pathway by activating adiponectin receptors with conditioned media from Adipoq^+/+^ cells that are known to secrete adiponectin. Conditioned media from Adipoq^+/+^ and adipoq^−/−^ cells showed a similar secretory profile. Silver staining detected a band around 30 kDa that was present in Adipoq^+/+^ cells and was missing in the adipoq^−/−^ sample (Figure 7A, arrow). Western blot of the same samples confirmed that this band was adiponectin. Treatment of adipoq^−/−^ cells with conditioned media from Adipoq^+/+^ cells showed reductions in AKT and MAPK1 phosphorylation as well as CCND1 and no changes in ACTA2 (Figure 7B,C).

Adiponectin has been shown to activate T-cadherin and AR1/AR2 in both mouse and human cell lines [33,34]. RNA sequence data showed that AR1 and AR2 may be upregulated, while the gene coding for T-cadherin (Cdh13) may be downregulated in adiponectin-deficient cells (Appendix A). To investigate whether activation of AR1 and AR2 mediates the effects of adiponectin in cell signaling, we used AdipoRon, an AR1 and AR2 agonist. AdipoRon increased the phosphorylation of the AMPK substrate acetyl-CoA carboxylase (ACC) and reduced AKT and MAPK1 phosphorylation and CCND1 in a concentration-dependent manner without affecting ACTA2 expression (Figure 7D,F,H). Further, activation of AMPK with metformin mimicked the effects of AdipoRon (Figure 7E,G,I). Altogether, these data suggest that adiponectin secreted by VSMCs activates AR1 and AR2, which mediates the downregulation of AKT and MAPK1 signaling and CCND1 in an AMPK-dependent manner. However, it is unknown why the reduction in AKT and MAPK1 activities by AdipoRon and metformin are unable to upregulate ACTA2.

### 3.4. AKT Mediates the Proliferation and Migration of Adipoq^−/−^ Cells

The basal rate of proliferation (Figure 3A) was significantly higher in adipoq^−/−^ cells compared with Adipoq^+/+^ cells. Inhibition of AKT, MAPK1, and TGF-β and activation of AMPK (AdipoRon and metformin) reduced CCND1 expression, suggesting that these treatments should also reduce proliferation of adipoq^−/−^ cells. Adipoq^+/+^ and adipoq^−/−^ cells were counted at basal (BS) and after 4 days of treatment with control media, media containing inhibitors of the AKT/MAPK1/TGF-β pathways, or media containing activators of adiponectin signaling (Figure 8A). The strongest reduction in proliferation was seen in cells treated with TCB, followed by AdipoRon and SB, all reaching cell numbers lower than Adipoq^+/+^ cells. Metformin reduced proliferation to levels like Adipoq^+/+^ cells. Inhibition of MAPK1 and EGFR led to reductions in cell numbers that did not reach significance. Like their effects on proliferation, TCB and SB treatment had the strongest effect on reducing migration of adipoq^−/−^ cells to levels lower than Adipoq^+/+^ cells (Figure 8B). Metformin and AdipoRon showed similar reductions in migration, consistent with Adipoq^+/+^ levels. Further, although PD reduced migration of adipoq^−/−^ cells, migration was still higher in comparison with Adipoq^+/+^ cells. Inhibition of the EGFR by Erl showed no effect on migration. Altogether, these data suggest that the upregulation of AKT signaling is the major contributor to the elevated rate of proliferation and migration of adipoq^−/−^ cells.

### 3.5. Loss of Adiponectin Dysregulates Cellular Signaling in Male and Female Aortas

To assess the significance of the molecular changes observed in vitro, we measured the expression of proteins of interest in the aortas of male (n = 8 per genotype) and female (n = 7 per genotype) Adipoq^+/+^ and adipoq^−/−^ mice (Figure 9). Aortas of adipoq^−/−^ males showed significant increases in the phosphorylation of AMPK, AKT, MAP2K1, and MAPK1 (Figure 7A) and expression of MMP2 and MMP3. No differences were seen for TAGLN or DCN. Aortas of adipoq^−/−^ females showed a robust upregulation of AMPK phosphorylation and expression of the EGFR, TAGLN, and MMP3 and a reduction in DCN (Figure 7B). Thus, the protein expression profile identified in VSMCs in vitro by adiponectin deficiency is relevant in vivo.

## 4. Discussion

RNA sequence data of male adipoq^−/−^ VSMCs furthered our understanding of the importance of adiponectin in cellular homeostasis while indicating potential key pathways involved in the regulation of the phenotypic switch of VSMCs. Like previous reports [5,35], adipoq^−/−^ cells underwent a switch to the synthetic phenotype with increased proliferation and migration and reduced expression of differentiation markers. This phenotype was associated with heightened expression of glycolytic enzymes at the RNA and protein levels and impaired OXPHOS. Upregulation of LDH suggests that the conversion of pyruvate into lactate provides a source of NAD needed to maintain a higher glycolytic rate. Analysis of pyruvate metabolism from the RNA sequence data suggests that the activity of the pyruvate dehydrogenase (PDH) complex should be upregulated in adipoq^−/−^ cells since expression of the pyruvate dehydrogenase kinase (PDK2 and 4), an inhibitor of the PDH complex, is downregulated, while the pyruvate dehydrogenase phosphatase (PDP2), an activator of the PDH complex, is upregulated. This suggests that Acetyl-CoA, a central regulator of epigenetic modifications, should be increased. Thus, a larger Acetyl-CoA pool in adiponectin deficient cells may promote the upregulation of glycolytic enzymes as well as other genes increased in these cells. Measurements of nicotinamide adenine dinucleotide (NAD) and Acetyl-CoA should be evaluated in future research to elucidate the role of adiponectin in epigenetic modifications in VSMCs.

Network analysis revealed a close interaction between the PI3K-AKT, ECM-receptor interaction, focal adhesion, and pathways in cancer in the adiponectin-deficient cells. We focused on the PI3K/AKT and ECM pathways to identify molecular mechanisms regulating the phenotypic switch, which is linked to vascular dysfunction and diseases like neointima formation and atherosclerosis.

Our in vitro data show that there are significant differences in the secretory profile, proliferation, migration, and differentiation with the loss of adiponectin. The signaling cascade through AKT, TGF-β, and AMPK are major potential targets for reversing the synthetic phenotype of adipoq^−/−^ VSMCs. Our data are consistent with a model (Figure 10) in which adiponectin activates AMPK, leading to reduced AKT and MAPK1 activation. Through the inhibition of AKT, adiponectin reduces the proliferation and migration of VSMCs, and through the inhibition of MAPK1, adiponectin maintains the differentiated state of VSMCs. In adipoq^−/−^ VSMCs, TGF-β promoted proliferation and migration through AKT activation. Our data agree with the current literature, which shows that TGF-β promotes AKT signaling and that is, as previously mentioned, a cause for increased VSMC proliferation [36]. Thus, it is likely that in adiponectin-deficient cells, inhibiting TGF-β reduces proliferation by an AKT-dependent mechanism. Additionally, we find that activating AMPK with AdipoRon produces similar effects to AKT and TGF-β inhibition. This is a significant finding because it agrees with previous reports on AdipoRon [10]. Also, in agreement with our data, Zhou et al. have shown that globular adiponectin inhibits osteoblastic differentiation of VSMCs through the PI3K/AKT and Wnt/β-catenin pathways [10]. This points back to the dysregulation of AKT signaling as the likely primary cause behind the increased proliferation and migration of adipoq^−/−^ VSMCs, which we have shown in this paper.

Our data agree with the work by Ding et al. that showed that adiponectin influences differentiation in human coronary artery smooth muscle cells [5]. Initial analysis of differentiation markers in adipoq^−/−^ cells showed a significant reduction in key proteins such as MHY11, ACTA2, TAGLN, and MYL. AKT induces MYOCD transcription by phosphorylation and inhibition of Foxo3a, a negative regulator of MYOCD transcription [37]. Thus, increased AKT activity may explain why MYOCD expression is upregulated. MYOCD and SRF upregulation could also be a compensatory mechanism to correct for the reduced expression of differentiation markers. The reduced effect of MYOCD and SRF suggests that promoter regions of differentiation genes are not accessible (closed chromatin) and/or that these transcription regulators are in the wrong compartment (cytosol instead of the nucleus). We observed a reduction in SRF nuclear localization in adiponectin-deficient cells that may explain, in part, the reduced expression of differentiation markers.

Inhibition of the activity of AKT by TCB and MAPK1 by PD induced a small but significant increase in ACTA2 and TAGLN expression. However, inhibition of these kinases via activation of adiponectin signaling with AdipoRon and metformin showed no effect on ACTA2 but reduced proliferation and migration of adipoq^−/−^ cells. Fairaq et al. demonstrated that AdipoRon reduced proliferation in an mTOR-dependent mechanism [10]. The effect of AKT in differentiation may be isoform specific. AKT1 and AKT2 have opposite effects on proliferation and differentiation. Knockdown of AKT1 induces a contractile phenotype, while knockdown of AKT2 induces a dedifferentiated state [38]. TGF-β is known to upregulate the phosphorylation of AKT2 to stimulate differentiation [5,35]. While TCB inhibits all AKT isoforms, it is unknown which AKT isoform is inhibited by the adiponectin/AMPK pathway. It is also possible that the phenotypic switch has more than one arm of regulation. Proliferation may be most impacted by AKT1, while differentiation may be most sensitive to a combination of both AKT2 and MAPK1. This is in line with the current literature, which has indicated that differentiation is regulated by MAPK [39]. The MAPK pathway regulates differentiation by phosphorylating (inhibiting) the MYOCD transactivation domain, which leads to a reduction in proteins like TAGLN and ACTA2 [40]. Also, our finding that AKT is implicated in differentiation, albeit not as strongly as MAPK, agrees with the current literature [41]. It is also possible that differentiation is mediated through T-cadherin. If T-cadherin is downregulated, as suggested by the RNA sequence, then conditioned media would fail to promote differentiation of VSMCs. In this scenario, activation of AR1 and AR2 by adiponectin may reduce AKT1 activity to reduce proliferation and migration, while activation of T-cadherin by adiponectin or TGF-β signaling may activate AKT2 to promote differentiation.

Concerning the ECM, molecules regulating TGF-β function were upregulated by adiponectin deficiency, including DCN, MMP14, and TFRC. The TGF-β pathway was more active in adipoq^−/−^ cells (increased AKT and SMAD2 phosphorylation), inducing a stronger upregulation of TFRC with TGF-β treatment. However, the upregulation of ACTA and MMP14 in response to TGF-β treatment in Adipoq^+/+^ cells was not observed in adipoq^−/−^ cells, suggesting that some arms of the TGF-β signaling pathway were impaired. Our data corroborate other studies that also determine that inhibition of TGF-β is detrimental to differentiation [42].

MMP2 and MMP9 were reduced, and MMP3 was upregulated in vitro. Our data agree with some findings that have shown adiponectin reducing MMP9 [43] but contradict the results of studies by Wanninger et al. that reported that adiponectin increases MMP9 activity. Notably, however, these studies were conducted in hepatocytes [44]. Consistent with our in vitro work, the phosphorylation of AKT, MAPK1, and MAP2K1, as well as the expression of MMP2 and MMP3 were upregulated in the aorta of adipoq^−/−^ male mice compared with Adipoq^+/+^. The upregulation of MMP2 and MMP3 was particularly interesting because of its known role in advancing atherosclerotic plaque development [45]. Only MPK2K1 phosphorylation and EGFR and MMP3 expression were elevated in females by adiponectin deficiency. The major sex-specific differences in vivo were for MMP3 and DCN, which showed a robust upregulation and downregulation, respectively, in females compared with males. Unexpectedly, AMPK activity was upregulated in both sexes by adiponectin deficiency, which may signify a compensatory mechanism in young mice. Along this line, Caldwell et al. uncovered a compensatory mechanism that protects young sedentary adiponectin-deficient male mice from cardiac and coronary microvascular dysfunction [46]. Females were not investigated in this study. The role of aging in this compensatory mechanism is unknown. The robust upregulation of MMP3 and downregulation of DCN likely impaired this compensatory mechanism in females. The sex-dependent effects of MMP3 and DCN in plaque require further elucidation in future research.

In conclusion, we showed that adiponectin maintains the contractile VSMC phenotype by inhibiting AKT-dependent proliferation and migration, MAPK-dependent dedifferentiation, and TGF-β-dependent ECM regulation. We also provided novel in vivo evidence of sex-specific differences in the expression of pro-atherogenic molecules in the aorta by adiponectin deficiency that may indicate sex-specific effects on atherosclerosis.

Some limitations of our study include (1) only cells derived from male mice were used to generate the RNA sequence and the in vitro data, (2) metabolites like NAD and Acetyl-CoA should be evaluated to identify the role of adiponectin in epigenetics in future research, and (3) only young mice were used to assess protein expression in the aorta.

## Figures and Tables

**Figure 1 cells-12-02493-f001:**
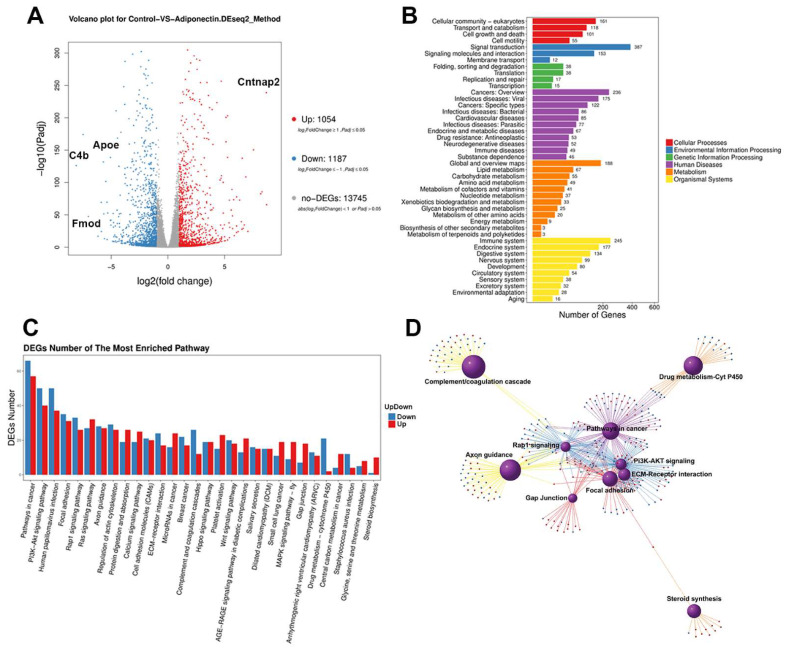
RNA sequence analysis of adiponectin deficient VSMCs. RNA sequence data of male Adipoq^+/+^ and adipoq^−/−^ VSMCs (n = 4 samples per genotype) are presented as a volcano plot of the gene distribution based on the log2 (adipoq^−/−^/Adipoq^+/+^) for RNA expression (**A**). Log2 (fold changes) >1 were upregulated (red) and <1 were downregulated (blue) genes. Changes in cellular processes (**B**) and DEGs of the most enriched pathways (**C**) show strong changes in signal transductions, in particular the PI3K/AKT pathway. Molecular interactions of the most altered pathways are shown in (**D**).

**Figure 2 cells-12-02493-f002:**
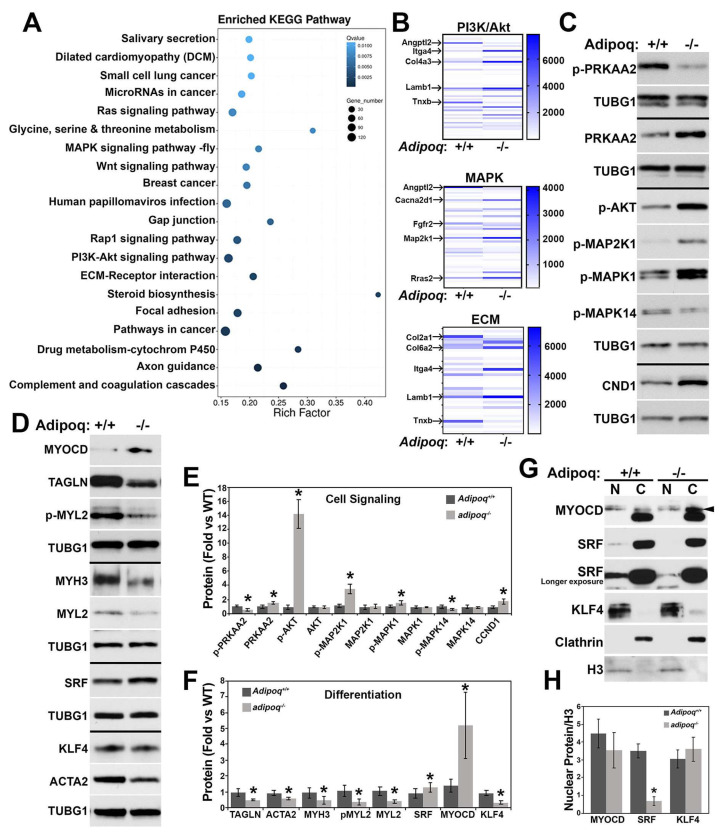
Vascular smooth muscle cell phenotype is altered by loss of adiponectin. RNA sequence data (n = 4/genotype) results show the most enriched KEGG pathways in the adipoq^−/−^ cells (**A**) and heat maps indicate the specific genes altered in the PI3K/AKT, MAPK, and ECM pathways (**B**). Confluent Adipoq^+/+^ and adipo*q*^−/−^ cells (n = 4–6) were starved in 0.2% FBS for 24 h and total extracts (30–50 μg) were tested for signaling cascade (**C**) and differentiation markers (**D**). Respective quantification graphs are shown in (**E**,**F**). Nuclear and cytosolic fractions (20 μg, n = 3) were tested for MYOCD, SRF, and KLF4 (**G**,**H**). * denotes *p* < 0.05 compared with Adipoq^+/+^.

**Figure 3 cells-12-02493-f003:**
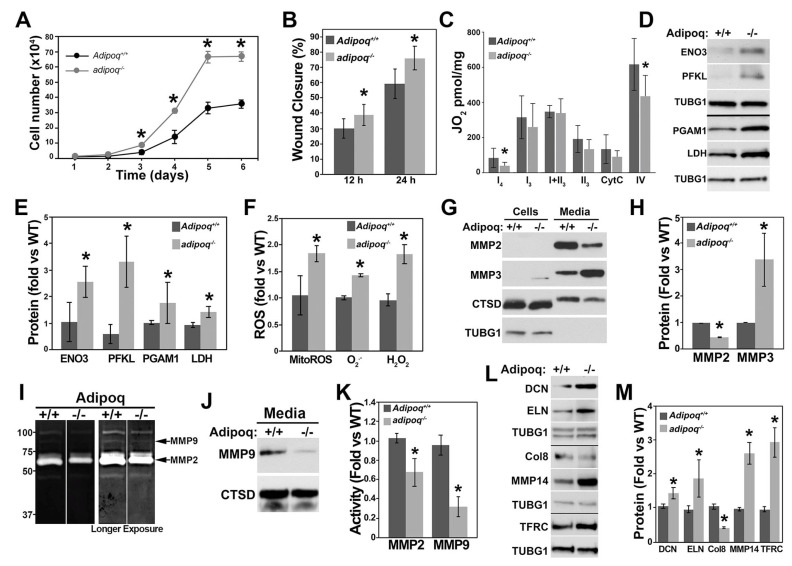
Loss of adiponectin alters metabolic activity, mitochondrial function, and ECM regulation. Adipoq^+/+^ and adipo*q*^−/−^ cells were counted every day for 6 days to assess the proliferation rate (n = 3–4 wells per experiment) (**A**). Migration was assessed by scratch wound closure assay (**B**). Oxygen flux (JO_2_) was assessed by high-resolution respirometry using the Oroboros O2k oxygraph. Complex I, state 2 (I_2_) respiration, and state 3 respiration rates of Complex I (I_3_), combined Complex I and Complex II (I + II_3_), Complex II (II_3_), and Complex IV (IV_3_) were assessed (**C**). Adipoq^+/+^ and adipoq^−/−^ cells were cultured to confluency and then starved for 24 h in 0.2% FBS for Western blot analysis of glycolytic enzymes (n = 3–4) (**D**,**E**) and ROS levels (**F**). Adipoq^+/+^ and adipoq^−/−^ cells were starved for 24 h in FBS-free media for analysis of membrane-bound and secreted MMPS (n = 3–4) (**G**,**H**). The media were also used for gelatin zymography analysis of MMP activity (**I**). Western blot analysis of the same media was performed to analyze the expression of MMPs (**J,K**). Expression of ECM proteins and TFRC in total extracts of Adipoq^+/+^ and adipoq^−/−^ cells (**L**,**M**). * denotes *p* < 0.05 compared with Adipoq^+/+^.

**Figure 4 cells-12-02493-f004:**
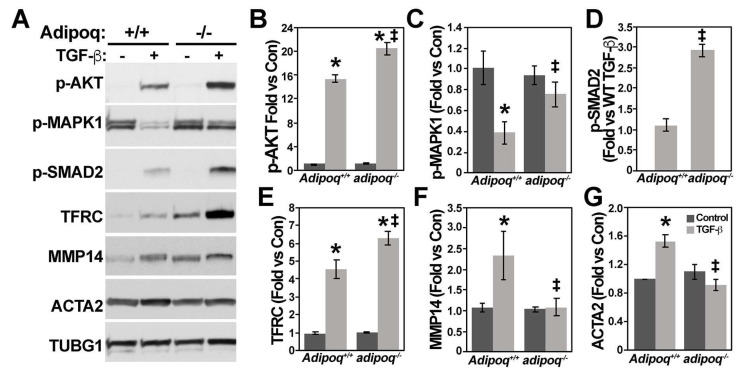
Altered cell signaling prevents TGF-β-induced differentiation of VSMCs in the absence of adiponectin. *Adipoq^+/+^* and *adipoq ^−/−^* cells (n = 3) starved for 24 h with 0.2% FBS were treated with 10 ng/mL TGF-β for 24 h to test the expression of in cell signaling, ECM regulation, and differentiation markers (**A**–**G**). * denotes *p* ≤ 0.05 compared to control, ‡ denotes *p* ≤ 0.05 compared to TGF-β treated *adipoq^-/-^* cells.

**Figure 5 cells-12-02493-f005:**
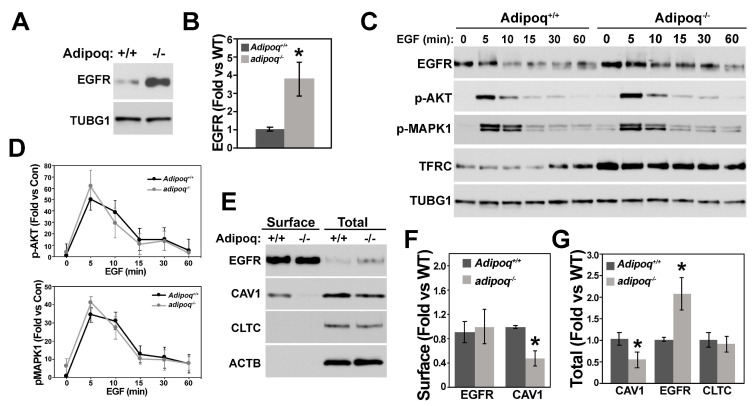
EGFR does not regulate the upregulation of AKT signaling by adiponectin deficiency. Adipoq^+/+^ and adipoq ^−/−^ VSMCs (n = 4) were starved for 24 h with 0.2% FBS EGFR expression (**A**,**B**) or were treated with 100 nM EGF for 0–60 m and total cell extracts probed for ERFR, AKT and MAPK1 phosphorylation, and TFRC expression (**C**). Quantification of phosphorylated kinases is shown in (**D**). Biotinylation shows changes in cell surface and total levels of EGFR, CAV-1, and TFRC (**E**–**G**). * denotes *p* < 0.05 compared with Adipoq^+/+^ controls. N = 3 independent experiments.

**Figure 6 cells-12-02493-f006:**
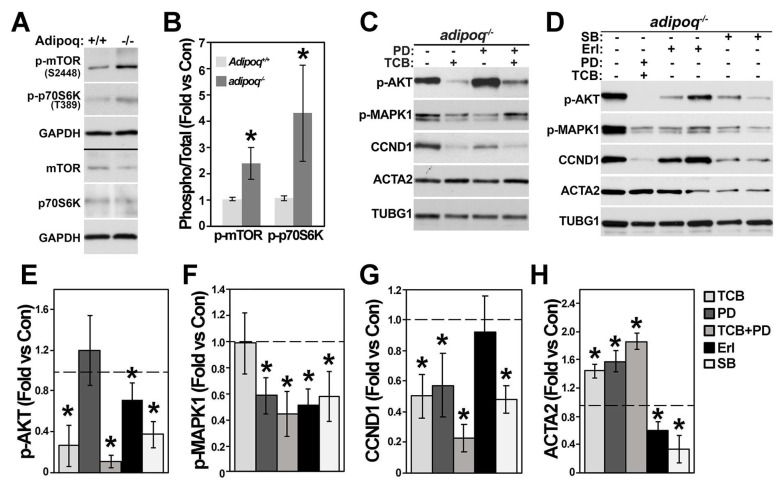
AKT and MAPK1 regulate VSMC phenotype. Adipoq^+/+^ and adipoq^−/−^ VSMCs were starved for 24 h with 0.2% FBS to measure mTOR phosphorylation and its downstream target (**A**,**B**) or were treated with 10 μM TCB and PD-98059 (**C**), 100 nM Erl, or 5 μM SB-252334 (**D**) for 24 h and total cell extracts used for Western blot. Target genes were adjusted by GADPH and then the ratio of phosphorylated protein was divided by total expression and expressed as fold change compared with Adipoq^+/+^ cells (**B**). Quantification of proteins in (**C**,**D**) was expressed as fold change compared with control (dotted lines) (**E**–**H**). Asterisks (*) represent *p* < 0.05 compared with adipoq^−/−^ control cells. (n = 3–4 independent experiments.)

**Figure 7 cells-12-02493-f007:**
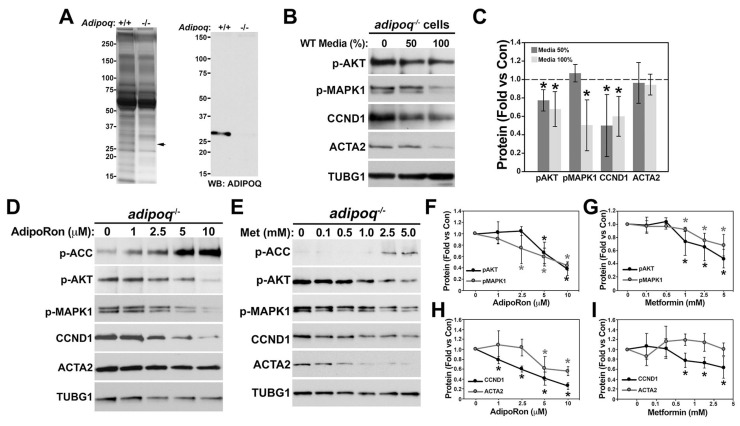
Restoration of adiponectin and AMPK activity reduces cell signaling but does not improve differentiation. Concentrated media from Adipoq^+/+^ and adipoq^−/−^ cells were separated in a 4–20% pre-cast Criterion gel and stained with silver or used for Western blot to measure adiponectin expression (**A**). Adipoq^−/−^ cells were treated for 48 h with and without undiluted Adipoq^+/+^ media (100%) or diluted 50% in plain media (**B**). Quantification for (**B**) is shown in (**C**). Adipoq^−/−^ cells were treated for 72 h with 1–10 μM AdipoRon (**D**,**F**,**H**) or 0.1–5 mM metformin (**E**,**G**,**I**). Quantification is expressed as fold change compared with control (dotted lines in (**D**)). * represents *p* < 0.05 compared with adipoq^−/−^ control cells. (n = 3–4 independent experiments.)

**Figure 8 cells-12-02493-f008:**
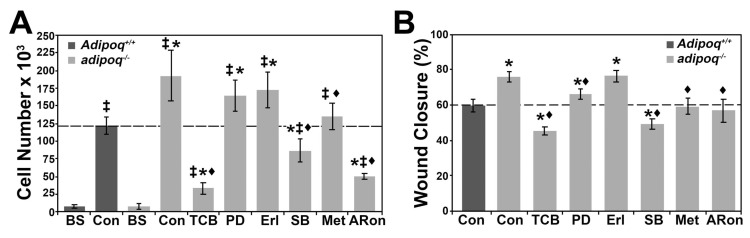
AKT is the major regulator of proliferation and migration in adiponectin-deficient cells. Adipoq^−/−^ cells were treated with 10 μM TCB, 10 μM PD-98059, 100 nM Erl, 5 μM SB-252334, 5 mM metformin, or 10 μM AdipoRon for 4 days to measure proliferation (**A**) or for 1 d in media with 0.2% FBS to measure migration (**B**). Cells were seeded at low confluency, starved for 48 h, and counted to establish basal (BS) cell numbers before treatment (**A**). Asterisks (*) represent *p* < 0.05 compared with Adipoq^+/+^ control, ‡ compared with BS for each genotype, and ♦ compared with adipoq^−/−^ control. N = 3 independent experiments.

**Figure 9 cells-12-02493-f009:**
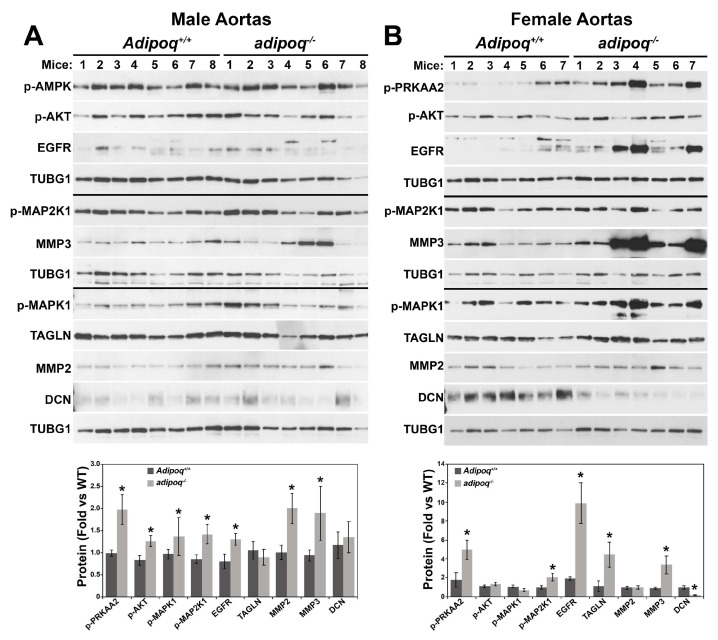
Sex-dependent effects in the expression of cell signaling and ECM markers in aortas of adiponectin-deficient mice. Extracts (20–30 μg/well) of aortas of 5-month-old male (**A**) and female (**B**) Adipoq^+/+^ and adipoq^−/−^ mice were separated in 4–20% pre-cast Criterion gels to evaluate changes in protein expression of cell signaling, ECM regulation, and differentiation molecules. * denotes *p* < 0.05 compared with Adipoq^+/+^ controls. N = 8 and 7 males and females per genotype, respectively.

**Figure 10 cells-12-02493-f010:**
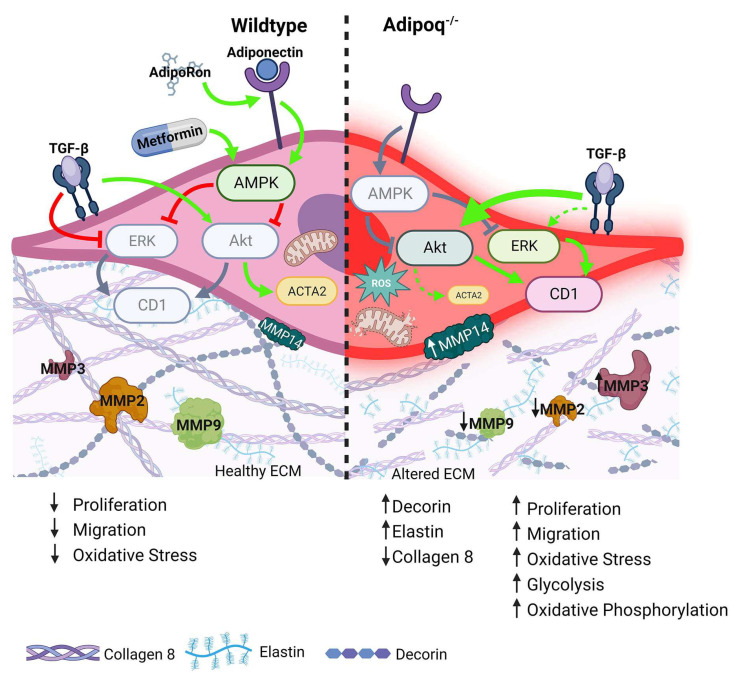
Model of the role of adiponectin signaling in VSMC phenotype. Adiponectin regulates the contractile phenotype of VSMC by activating the AMPK pathway, which reduces AKT and MAPK1 activity, leading to reduced CCND1 (CD1). Overall, this pathway promotes the expression of contractile genes, reducing proliferation, migration, and oxidative stress. In the absence of adiponectin, the lack of AMPK activity allows AKT and MAPK1 activation/phosphorylation and downstream signaling pathways that activate proliferation, migration, and oxidative stress. Significant changes are also seen in the ECM as decorin and elastin are upregulated, while collagen 8 is reduced in the absence of adiponectin. The figure was generated using BioRender.com (accessed on 5 October 2023).

## Data Availability

The data presented in this study for RNA sequence is available upon request.

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
