# Peer review of "AKT Mediates Adiponectin-Dependent Regulation of VSMC Phenotype"

_cells, 2023, doi:10.3390/cells12202493_

Round 1
Reviewer 1 Report
Manuscript entitled “AKT mediates adiponectin-dependent regulation of VSMC phenotype. Number: cells-2643754
An interesting manuscript about the signaling mediated by AKT on VSMC phenotype regulation.
The work is worthy of publication
Minor point
In legend of Figure 8; TCB instead of TBC

Author Response
Response to previous review
Reviewer 1
An interesting manuscript about the signaling mediated by AKT on VSMC phenotype regulation.
The work is worthy of publication.
Minor point
In legend of Figure 8; TCB instead of TBC.
Response:
Thank you for considering our work worth of publication. We have revised the legend for Figure 8.
Reviewer 2 Report
The manuscript by Cullen AE and colleagues describes the effects of adiponectin gene knockout in the VSMC phenotype. Data showed that adipoc -/- VSMC from male aortas showed increased Akt phosphorylation associate with increased cell migration. These cells showed alteration in energy metabolism. Also, they showed sex differences effects.
However, some major concerns should be addressed:
1. This is a descriptive work showing the effects of the global adiponectin gene knockout on VSMC. Only in figure 6 the use of Akt and MEK inhibitors were used to evaluate CCND1 and ACTA2 levels. However, these data are insufficient to conclude that both signaling are involved in the control of the phenotypic switch to a synthetic phenotype. More robust intervention and a more complete panel of evaluation is required.
2. In some experiments the number of replicas is missing. For example, in figures 2, 4, 8, 9.
3. In figure 2, the authors Akt and Erk are activated, and AMPK is inhibited because its phosphorylation levels. No experiments were performed measuring downstream targets to verify that these phosphorylation changes are indeed associated with changes on kinase activity. In figure 3E, Akt is displayed as Akt1. This should be corrected because the antibody is a pan-Akt.
4. Modified the adiponectin knockout the mTORc1 activity?
5. In figure 5A, Adipoq -/- cells showed lower growth rate than the Adipoq +/+ cells. In fat at time 2d and 3d, there are more Adipo +/+ cells than the adipoc -/- cells. These suggest that the adiponectin knockout reduce proliferation rate. This interpretation is contrary to that suggested by the authors. To corroborate authors information other complementary assays are required, for instance: cell cycle determination, cyclins levels, Ki67, etc.
6. Wound healing assay also is altered by cell proliferation, particularly because in this work this technique is performed at 12 and 24 h. Therefore, complementary assay, for instance by using transwell assay, is required.
7. In the manuscript, authors suggest that the adiponectin gene KO induce the switch toward the synthetic phenotype. However, no ECM protein synthesis and secretion, for instance collage and elastin, were measured.
8. It is not clear why the experiments in figure the activation of adiponectin receptor using recombinant adiponectin to adipoc -/- is not sufficient to inhibit Akt and MAPK1 phosphorylation like adipoc +/+ cells. A quantification of adiponectin receptors is recommended.
Author Response
Reviewer 2
The manuscript by Cullen AE and colleagues describes the effects of adiponectin gene knockout in the VSMC phenotype. Data showed that adipoc -/- VSMC from male aortas showed increased Akt phosphorylation associate with increased cell migration. These cells showed alteration in energy metabolism. Also, they showed sex differences effects.
However, some major concerns should be addressed:
- This is a descriptive work showing the effects of the global adiponectin gene knockout on VSMC. Only in figure 6 the use of Akt and MEK inhibitors were used to evaluate CCND1 and ACTA2 levels. However, these data are insufficient to conclude that both signaling are involved in the control of the phenotypic switch to a synthetic phenotype. More robust intervention and a more complete panel of evaluation is required.
Response:
To elucidate the role of AKT and MAPK pathways in the phenotypic switch we also evaluated the impact of these inhibitors in proliferation and migration in Figure 8. Elevated proliferation and migration are prominent phenotypes of synthetic VSMCs. AKT inhibition showed a strong effect in reducing these phenotypes, compared with MEK inhibition. We also added an additional marker of differentiation SM22/TAGLN (new Fig. S4), which strengthen our conclusion that both pathways regulate the phenotypic switch.
- In some experiments the number of replicas is missing. For example, in figures 2, 4, 8, 9.
Response:
The number of replicas was added to the figures, as suggested.
- In figure 2, the authors Akt and Erk are activated, and AMPK is inhibited because its phosphorylation levels. No experiments were performed measuring downstream targets to verify that these phosphorylation changes are indeed associated with changes on kinase activity. In figure 3E, Akt is displayed as Akt1. This should be corrected because the antibody is a pan-Akt.
Response:
Akt1 was corrected to Akt in Figure 2E. We also measured mTORC1 phosphorylation and its downstream targets, as mentioned in Q#4 below, to assess AKT activity. We were unable to measure activation of ERK targets. We order antibodies against phosphorylated p90 RSK, a well-kwon ERK target, however the order was delayed and will arrive by October 20, which is beyond the 10 days deadline for submission. We hope the reviewer will understand.
- Modified the adiponectin knockout the mTORc1 activity?
Response: Yes, we measured mTORC1 activity and its downstream targets and found that phosphorylation of mTOR, and p70SK were upregulated in adiponectin deficient compared with wild type cells.
- In figure 5A, Adipoq -/- cells showed lower growth rate than the Adipoq +/+ cells. In fat at time 2d and 3d, there are more Adipo +/+ cells than the adipoc -/- cells. These suggest that the adiponectin knockout reduce proliferation rate. This interpretation is contrary to that suggested by the authors. To corroborate authors information other complementary assays are required, for instance: cell cycle determination, cyclins levels, Ki67, etc.
Response:
The reviewer is right, the figure presented in the original submission is a poor representation of the actual phenotype. A different proliferation curve is now presented in Figure 3A. We also added a new supplemental figure (Fig. S3) showing images of cells cultured in media with 10%FBS. Since we started culturing these cells, we noticed more cells going through mitosis in the adiponectin deficient cells. Quantification of cells going through telophase and cytokinesis are shown in Fig. S3C.
- Wound healing assay also is altered by cell proliferation, particularly because in this work this technique is performed at 12 and 24 h. Therefore, complementary assay, for instance by using transwell assay, is required.
Response:
It is unlikely that proliferation would interfere with the migration assay since the incubation was done in media containing 0.2% FBS (starvation media). This information was added to the methods section and figure legend.
- In the manuscript, authors suggest that the adiponectin gene KO induce the switch toward the synthetic phenotype. However, no ECM protein synthesis and secretion, for instance collage and elastin, were measured.
Response:
In the original submission the expression of decorin (DCN), elastin (ELN) and collagen 8 (Col8) were shown in Fig. 3L.
- It is not clear why the experiments in figure the activation of adiponectin receptor using recombinant adiponectin to adipoq-/- is not sufficient to inhibit Akt and MAPK1 phosphorylation like adipoq+/+ cells. A quantification of adiponectin receptors is recommended.
Response
We saw a lot of variability in the response to recombinant adiponectin. We do not know why cells do not respond more strongly to the recombinant hormone. Adiponectin is found in several forms including trimers, hexamers and high molecular weight complexes. It is possible that the recombinant adiponectin in solution did not form the structure required to activate the receptors. Receptors were present because cells responded to AdipoRon by activating AMPK. Since this data cannot be explained, we decided to remove it from the paper.
Round 2
Reviewer 2 Report
The authors responded appropriately to all the comments